# Enhanced Electromagnetic Wave Absorption of SiOC/Porous Carbon Composites

**DOI:** 10.3390/ma15248864

**Published:** 2022-12-12

**Authors:** Wen Yang, Li Li, Yongzhao Hou, Yun Liu, Xinwei Xiao

**Affiliations:** 1School of Transportation and Vehicle Engineering, Shandong University of Technology, Zibo 255000, China; 2Shandong Si-Nano Materials Technology Co., Ltd., Zibo 255400, China; 3School of Materials Science and Engineering, Shandong University of Technology, Zibo 255000, China; 4Shandong Industrial Ceramics Research & Design Institute Co., Ltd., Zibo 255400, China

**Keywords:** porous carbon, SiOC spheres, microwave absorption

## Abstract

Carbon-based materials have been widely explored as electromagnetic (EM) wave absorbing materials with specific surface areas and low density. Herein, novel porous carbon/SiOC ceramic composites materials (porous C/sp-SiOC) were prepared from the binary mixture, which used the low cost pitch as carbon resource and the polysilylacetylene (PSA) as SiOC ceramic precursor. With the melt-blending-phase separation route, the PSA resin formed micro-spheres in the pitch. Then, numerous SiOC ceramic micro-spheres were generated in porous carbon matrices during the pyrolysis process. By changing the percent of SiOC, the microstructure and wave absorption of porous C/sp-SiOC composites could be adjusted. The synergistic effect of the unique structure, the strong interfacial polarization, and the optimized impedance matching properties contributed to the excellent absorption performance of porous C/sp-SiOC composites. The minimum reflection loss for porous C/sp-SiOC absorber reached −56.85 dB, and the widest effective bandwidth was more than 4 GHz with a thickness of only 1.39 mm. This presented research provides an innovative and practical approach to developing high-performance porous carbon-based microwave absorption materials from green chemistry.

## 1. Introduction

The emergence and broad application of electronic equipment (telephones, computers, base stations, etc.) have caused serious microwave radiation and microwave pollution, which is a serious threat to human health. Therefore, it is necessary to develop a novel absorbing material that can effectively absorb or shield electromagnetic (EM) waves [1,2]. The ideal absorbing material should have the characteristics of high absorbing wavelength, wide absorbing frequency band, light weight, good thermal stability, and high economic benefit [3].

Carbon fiber [4], graphene [5], and other carbon materials are ideal candidates for microwave absorbing materials because of their tunable dielectric properties, good composite properties, low density, and good chemical stability. In particular, due to its low density and large specific surface area [6], porous carbon materials such as graphene foams [7], hierarchically porous carbons [8], and mesoporous carbon hollow spheres [9] have attracted intensive attention from researchers. The existence of pores is conductive to the multiple reflections of incident electromagnetic waves. In addition, pores can also be used as polarization centers to further improve the dielectric loss ability through multiple relaxation processes [10]. However, the high permittivity caused by good conductivity makes the impedance matching between pure carbon materials and free air poor, and microwave reflection rather than microwave absorption can easily occur on the sample surface, which leads to poor microwave absorption performance of pure carbon materials [11,12]. 

Fortunately, the introduction of ceramic elements into pure carbon materials is beneficial to improve the electromagnetic absorption performance, such as SiC [13], Si-C-N [14], Si-B-C-N [15], etc. These composite materials can effectively adjust the electromagnetic parameters, balance the impedance matching conditions, and improve the electromagnetic wave absorption performance of materials. There are two ways to attach ceramic components: to the surface of carbon materials or embedded in carbon materials [16]. Ye et al. obtained porous carbon material by pyrolysis with melamine as a raw material. In order to realize the perfect combination of carbon fiber (CF) skeleton and silicon carbide film, a layer of silicon carbide film was deposited on the porous carbon skeleton by chemical vapor deposition (CVD) with methyktrichlorosilane as the SiC source. The results showed that the SiC coating significantly improved its micro absorption properties. The prepared SiC/PyC-coated CF sample had an RL_min_ value of −33.6 dB at 7.16 GHz, and the thickness was only 2.25 mm [17].

However, due to the poor adhesion between carbon and ceramic components, its use as microwave absorbing materials is greatly limited [13]. Therefore, the strategy of embedding the special structure of ceramics in carbon materials is easier to accept. Han et al. prepared three-dimensional graphene oxide (GO) foam by a freeze-drying method. The SiC nanowires were then grown in situ by heating the silicon and silica powders, and rGO/SiC NW foam was obtained. It was effectively absorbed in the whole X band, which was thinner (3 mm) compared with the pure reduced graphene oxide foam. The effective bandwidth in X-band was 4.2 mm, and the RL_min_ was −19.6 dB [18]. Wen et al. designed and synthesized Co_3_O_4_@C nanosheets using NaCl particles as a template and Co(NO_3_)_3_∙6H_2_O and glucose as raw materials through a heat treatment annealing process (750 °C −2 h Ar and 250 °C −6 h air). Structurally, the synergistic effect between Co_3_O_4_ nanoparticles and coated porous carbon nanoparticles can well match the microwave absorption. At 11.4 GHz, the RLmin value was −32.3 dB, and the matching thickness was 2.3 mm [19].

In this paper, a composite material with SiOC ceramic spheres embedded in a 3D porous carbon structure was designed and prepared through a polymer-derived method using pitch and PSA, which served as carbon and SiOC precursor, respectively [20,21]. This new absorbing material with a unique structure combines porous carbon and ceramic materials’ advantages and also has a relatively simple process. The microwave absorbing properties of porous C/sp-SiOC composites can be adjusted by the ratio of Pitch/PSA. The phase structure and micromorphology of the composite materials have been investigated in detail. The absorbing mechanisms behind the excellent microwave absorption capacity are also discussed. This work demonstrates that porous C/sp-SiOC composite materials with superior electromagnetic absorption performances can be considered one of EWA materials’ efficient candidates.

## 2. Materials and Methods

### 2.1. Materials

In this work, the pitch was purchased from Jining Carbon Group Co., Ltd. (Jining, China) (C: 92.97 wt%, S: 0.484 wt%, N: 0.86 wt%, O: 1.12 wt%, and H: 4.566 wt%), and the properties of raw pitch are listed in Appendix A. The polysilylacetylene (PSA) used in this work was synthesized in the lab, and the detailed process was described in previous work [20,21]. The density of liquid PSA is approximately 0.9 g/cm^3^, and the ceramic yield is 80% after pyrolysis at 1000 °C.

### 2.2. Synthesis of Porous C/sp-SiOC Composites

As shown in Figure 1, the pitch was heated to 100 °C in an oil bath pan to obtain molten liquid pitch, and then different proportions of PSA were added into the molten pitch. After stirring for two hours at 150 °C, the liquid mixture (PSA/pitch) was cooled down to room temperature. PSA/pitch powder was then transferred to a tube furnace and carbonized at 1000 °C for one hour in an inert atmosphere to ensure the complete carbonization of the composite. Finally, the black products were obtained by natural cooling to room temperature. The names of these samples are shown in Table 1.

### 2.3. Characterization

A scanning electron microscope (SEM, Zeiss Merlin compact scanning electron microscope) was used to test the microstructure of the samples, and the elements of the internal structure were tested by energy dispersive spectrometer (EDS, Bruker Quantax Xflash 60 SDD). The surface area and pore diameter measurements of porous C/sp-SiOC composites were measured by Brunner-Emmet-Teller (BET, Micromeritics ASAP) equipment with nitrogen gas adsorption–desorption at −196 °C. X-ray diffractometer (XRD, Rigaku XRD 2500, Cu Kα, 40.0 KV, 30.0 mA) was used to identify the phase structures of composite materials. The thermo-stability of samples was obtained using a thermogravimetric analysis (TG, PerkinElmer Diamond) apparatus with a temperature of 10 °C/min in a nitrogen atmosphere. The chemical structures of samples were obtained using a Fourier transform infrared spectrometer (FT-IR, Thermo Nicolet AVATAR 370) with a resolution of 0.2 cm^−1^ in the range of 400–4000 cm^−1^.

The samples used for electromagnetic microwave absorbing measurements were first prepared by a uniform mixture of SiC@C powders (50 wt%) and paraffin (50 wt%) at 100 °C. Then, the mixture was pressed into a ring with a 3.0 mm inner diameter and 7.0 mm outer diameter to make toroidal ring samples. Next, the relative permittivity of synthesized samples was measured in the 2–18 GHz frequency range by Agilent Technologies N5280A vector network to calculate reflection loss.

## 3. Results and Discussion

### 3.1. Characterizations of Raw Materials

The element analysis of raw pitch was carried out by an element analyzer (Vario EL Cube, Elementar, Germany), as shown in Appendix A. The results show that pitch is composed of the four elements of C, H, N, and S, of which the C element accounts for the majority, followed by the H element. The infrared spectrum of pitch and PSA resin is shown in Appendix A. It can be seen from the pitch spectrum that it contains a C-H bond and C=C bond, including stretching vibration [22], bending vibration of C-H bond [23], and the stretching absorption of C=C groups [24,25]. From the spectrum of PSA resin, it can be seen that PSA resin mainly includes -Si-H bond stretching vibration, C≡C stretching vibration, and Si-CH_3_ vibrations [20], which can be concluded is {−[SiHCH_3_-C≡C-]_n_}, whose molecular structure is shown in Appendix A. The details of the characteristic interval of pitch and PSA are shown in Appendix A.

The pyrolysis product of PSA resin at 1000 °C is SiOC ceramic, and the ceramic yield is 80% [26]. Figure 2 shows the TG curves of pitch/PSA-n samples. The residual products in the TG curve are attributed to C/SiOC. It can be seen from the figure that the carbon residue of pitch at 1000 °C is only 28.33%. With the increase in PSA content, the ceramic yield of pitch/PSA increases gradually, and the ceramic yield of pitch/PSA-12 reached 42.96%. According to the ceramic yield of PSA resin and the TG curve of porous C/sp-SiOC-n, the weight percentage of SiOC and C in the composite can be calculated, and the actual density of the powder can be measured by the specific gravity bottle method; the details are shown in Table 2. In contrast, the density of composite materials is low, which meets the lightweight requirements of absorbing materials.

### 3.2. The Microstructure of Porous C/sp-SiOC

Appendix A shows the macroscopic photographs of porous C/sp-SiOC-n (n = 0, 4, 6, 8, and 12) composites after carbonization at 1000 °C. When the powder of the same quality is pressed into the block with the same diameter and pyrolyzed in a tubular furnace at 1000 °C, the pitch as a whole exhibits a loose expansion structure and poor mechanical properties. In contrast, porous C/sp-SiOC-n (n = 4, 6, 8, and 12) composites have good formability and certain mechanical properties.

In order to better study morphology and internal structure of the samples, the block of the porous C/sp-SiOC composites was analyzed by SEM, as shown in Figure 3. Due to the poor self-formability and loose structure of porous C/sp-SiOC-0 composite, the results show that there is no mesoporous material (shown in Figure 3a). The microstructure of the porous C/sp-SiOC-n (4, 6, 8, 12) composites shows an adequate intercommunicating porous structure. The pore size shows a trend of first increasing and then decreasing, which is basically maintained in the range of 40–150 μm, in which the pore size of porous C/sp-SiOC-8 is the smallest. In addition, it was also observed that SiOC particles with a diameter of 1–6 μm were embedded into the porous skeleton and inner wall of the material, and the number of spheres gradually increased with the increase in PSA content. With the further increase in PSA, SiOC particles formed obvious sintering necks [27]. The sintering neck is indicated by the arrows (as shown in Appendix A). EDS mapping confirmed that the particles were SiOC, as shown in Figure 4. The results show that the position of spherical particles is obvious, the content of Si and O elements is high (as shown in Appendix A), and the C element is evenly distributed in the whole matrix.

During the preparation process, PSA was partially cross-linked with pitch. According to the Plateau–Rayleigh instability theory, there is large interface energy between liquid PSA resin and coal tar pitch, and liquid PSA exists in the form of spherical droplets in the preparation process. During the pyrolysis process, pitch is transformed into carbon. At the same time, PSA is transformed into SiOC particles with a certain volume shrinkage, and the surface molecules have a large tensile force to receive ceramic molecules and a small tensile force to receive pitch carbon molecules (because of the low density of carbon), so the surface molecules are pulled into the bulk phase. With the increase in PSA content, SiOC particles formed in the carbon matrix also gradually increase, and the tensile force also gradually increases. Therefore, the pore size can be adjusted by the content of PSA.

Figure 5 shows the N_2_ adsorption–desorption isotherms and pore size distributions of the synthesized samples. All samples have the feature of IV-type curves with H3 hysteresis, indicating a typical characteristic of mesopores [28]. At high relative pressure (P/P0 > 0.9), nitrogen amounts absorbed rose steeply, indicating that these mesopores were not uniform, and some macropores were also present. The detailed data of the specific surface area and pore size of all samples are shown in Table 3. As the PSA content increased, the specific surface area increased first and then remained basically unchanged, and the pore size decreased slightly and then increased, which was consistent with the SEM results. Among them, the maximum specific surface area of porous C/sp-SiOC-8 composite was 7.6335 m^2^g^−1^, and the pore size was the smallest, concentrated around 7.9 nm. Details of other samples are shown in Appendix A.

Figure 6a shows the XRD patterns of all the porous C/sp-SiOC samples. It can be seen that a broad diffraction peak occurred at roughly 25°, being ascribed to the amorphous carbon component [29]. When the PSA content increased to a certain value, the diffraction peak at 2θ=26.5° appeared, indicating that the ordered carbon structure increased. This may be because the porous C/sp-SiOC-8 composite has the largest specific surface area, which makes the contact between pitch and PSA resin very wide, prone to more cross-linking, forming more C-C bonds, and showing a better-ordered structure. This discussion still needs further verification.

The graphitization degree and defects in porous C/sp-SiOC composite materials can further be investigated by Raman analysis. As illustrated in Figure 6b, the Raman spectra of these samples display quite similar spectra with two different peaks: the D band centered at ca. 1350 cm^−1^ and the G band at ca. 1590 cm^−1^ [30,31,32]. The G band is associated with sp2- hybridized carbon bonds, whereas the D band is linked to a disordered and defective carbon structure [33,34]. Therefore, I_D_/I_G_ can become a graphitization degree index [35,36,37]. Table 4 shows the ID/IG values for all samples. In contrast, the ratio of I_D_/I_G_ of porous C/sp-SiOC-n (n = 4, 6, 8, and 12) composite materials is less than that of the porous C/sp-SiOC-0 composites. In addition, it was found that the ID/IG ratio for porous C/sp-SiOC-8 composites was the smallest, which is associated with the formation of microcrystalline graphite.

The size of graphite microcrystalline is calculated $L_{a}=4.4\;\text{×}\;{\left(\frac{\mathrm{ID}}{\mathrm{IG}}\right)}^{-1}$, and the detailed dimensions of graphite crystallites are summarized in Table 4. The results of the samples are similar, and the calculated values are in the range 4.11–4.49. They all show a trend of first increasing and then decreasing. The porous C/sp-SiOC-8 sample is the largest, which is 4.63. This reveals that, in the XRD spectrum, only graphite microcrystalline peak is detected in the porous C/sp-SiOC-8 sample, and the others are amorphous C/SiOC peaks.

In summary, PSA and pitch were partially cross-linked during the preparation process. Liquid PSA exists in pitch in the form of spherical droplets induced by the interface energy. After pyrolysis at 1000 °C, PSA was pyrolyzed to SiOC particles, and pitch was pyrolyzed to carbon matrix. Due to the good adhesion of the two and the volume shrinkage of PSA resin during curing and cracking, a special structure of SiOC ceramic particles embedded in porous carbon material was formed. It has good formability, and the product has certain mechanical properties. With the increase in PSA content, the ceramic yield of pitch/PSA increased gradually. By changing the amount of PSA, the number of SiOC ceramics and specific surface area and pore size of porous C/sp-SiOC composites can be adjusted. With the increase in PSA content, the number of SiOC ceramic microspheres in porous carbon composites can be adjusted. With the increase in PSA content, the number of SiOC ceramic microspheres increases, the specific surface area of the composite first increases and then decreases, and the pore size and the number of defects first decrease and then increase.

### 3.3. The Electromagnetic Properties of Porous C/sp-SiOC

The ideal electromagnetic materials should have the characteristics of being thin, lightweight, and broadband, with strong adsorption capacity. The reflection loss (RL) curves for these as-prepared samples (50 wt% mixed with paraffin) are displayed in Figure 7. In Figure 7a, the porous C/sp-SiOC-0 composite showed a weak absorption, with the RL value from 0 dB to −7 dB. However, porous C/sp-SiOC-n (n = 4, 6, 8, and 12) materials all showed enhanced microwave absorption properties in contrast to the porous C/sp-SiOC-0 composite. From the 3D EM plots, the RL_min_ value of porous C/sp-SiOC-4 was −16.6 dB at 13.8 GHz at a thickness of 1.5 mm, the RL_min_ of porous C/sp-SiOC-6 composite at 2.0 mm reached −43.46 dB at 10.8 GHz, the RL_min_ of porous C/sp-SiOC-8 was −56.85 dB at 16 GHz with a thickness of only 1.39 mm, and the RL_min_ of porous C/sp-SiOC-12 composite reached −72.25 dB with a thickness of 4.57 mm.

In addition, the microwave absorption properties of samples at different thicknesses exhibit obvious changes and are clearly reflected in the results. Under the same thickness, the absorption bands of different samples remain unchanged. Due to the quarter-wave-length attenuation theory, all samples trended toward the lower frequency with the increase in thickness, as described in Equation (1).
(1)$$t_{m}=\frac{n\lambda }{4}$$
where *t_m_* stands for the thickness of absorbers, λ is the wavelength, and n is taken as the odd number (1, 3, 5……). With the increase in PSA content, the overall bandwidth of porous composite samples increased first and then decreased, taking RL_min_ = −15 dB and a thickness of 1.5 mm as an example. Porous C/sp-SiOC-8 had the broadest effective absorption bandwidth (EAB), reaching 4.3 GHz (12.7 GHz–17 GHz). The EW in X and Ku bands was achieved by adjusting the absorber thickness from 1.3 mm to 3.0 mm. As illustrated in Appendix A, when the matching thickness changes from 1.23 mm to 1.55 mm, the reflection loss below −20 dB covers the Ku band, and the absorption bandwidth can reach 4–4.33 GHz.

The electromagnetic absorption characteristics are intimately related to the complex permittivity (*ε_r_*) and permeability (*μ_r_*). The *μ_r_* is not considered because porous C/sp-SiOC composites are non-magnetic materials, so only the *ε_r_* is shown in Figure 8. The real parts of complex permittivity (*ε′*) represent the storage capacity of EM wave energy, and the imaginary parts (*ε*″) stand for the dielectric loss ability [38]. To better understand the absorption performance of EMW materials, the relative complex permittivity (Equation (2)) was recorded in the range of 2–18 GHz.
(2)$$\varepsilon_{r}=\varepsilon^{\prime }-j\varepsilon^{\prime \prime }$$

All samples gradually maintained the overall downward trend with increasing frequency (Figure 8a,b), which shows a frequency dependence behavior. Compared with others, the measured dielectric constant of porous C/sp-SiOC-0 samples was very high, and the real and imaginary parts at 8 GHz reach 25 and 20, which are closely related to the carbon-based materials’ high electrical conductivity properties. However, high conductivity creates a lot of microwave reflection on the surface of the absorber, adversely affecting the electromagnetic wave absorption behavior and leading to poor impedance matching. For other composites, the unique structure of SiOC ceramic microspheres embedded in the porous carbon matrices hinders the migration of carriers between carbon, which results in low *ε′* and *ε*″ values, and both *ε′* and *ε*″ are stable between 25-10 and 15-3, respectively. For the porous C/sp-SiOC-n (n = 4, 6, 8, and 12) composites, the dielectric loss tangent (*tan*δ = *ε′*/*ε*″) is less than 0.9, meaning that the dielectric loss is a dielectric loss-dominated absorbing material (shown in Figure 8c).

To further reveal the inner mechanisms of their outstanding EM absorption performance, the dielectric behaviors of porous C/sp-SiOC-n (n = 0, 4, 6, 8, and 12) composite materials were systematically analyzed. The dielectric loss of samples has been studied using Debye dipolar relaxation [39]. According to the Debye theory, *ε_r_* can be expressed as follows:(3)$$\varepsilon_{r}=\varepsilon^{\prime }-j\varepsilon^{\prime \prime }=\varepsilon_{\infty }+\frac{\varepsilon_{s}-\varepsilon_{\infty }}{1+j2\pi f\tau }$$

The relative dielectric constant at the high-frequency limit is denoted by $\varepsilon_{\infty }$, whereas the static dielectric constant is denoted by *ε_s_*; *f* stands for the frequency, and τ refers to the polarization relaxation time. Hence, *ε′* and *ε*″ are also expressed as:(4)$$\varepsilon^{\prime }=\varepsilon_{\infty }+\frac{\varepsilon_{s}-\varepsilon_{\infty }}{1+{\left(2\pi f\right)}^{2}\tau^{2}}$$
(5)$$\varepsilon^{\prime \prime }=\frac{2\pi f\tau \left(\varepsilon_{s}-\varepsilon_{\infty }\right)}{1+{\left(2\pi f\right)}^{2}\tau^{2}}$$

Combining the two equations, $\varepsilon^{\prime }$ and $\varepsilon^{\prime \prime }$ could be depicted as:(6)$${\left(\varepsilon^{\prime }-\frac{\varepsilon_{s}+\varepsilon_{\infty }}{2}\right)}^{2}+{\varepsilon^{\prime \prime }}^{2}={\left(\frac{\varepsilon_{s}-\varepsilon_{\infty }}{2}\right)}^{2}$$

According to these equations, it can be described as a semicircle called the Cole–Cole semicircle. Each semicircle corresponds to a Debye relaxation process [40]. All samples except the porous C/sp-SiOC-0 sample have multiple semicircles, shown in Figure 9. The long tail of each semicircle represents the conducting loss process, and these semicircles reveal that multi-relaxation polarization occurs in the porous C/sp-SiOC-n (n = 4, 6, 8, and 12) composites, indicating that the Debye dipolar relaxation has a positive effect on the dielectric loss. Here, interfacial polarization and dipole polarization play important roles in the absorbing behavior of porous C/sp-SiOC composites. For the porous C/sp-SiOC-n (n = 4, 6, 8, and 12) composites, due to the interfacial interaction between amorphous SiOC spheres and pyrolytic carbon, the charge distribution on both sides of the interface is also different, which leads to the interfacial polarization [41]. In addition, defects in the carbon matrix also cause dipole relaxation to consume most of the energy of the electromagnetic wave [42].

Good impedance matching is a prerequisite condition for absorbers to exhibit excellent electromagnetic wave absorption performance [43]. The ratio Z_in_/Z_0_ = 1 indicates that the EM waves from the air could easily enter the composites rather than reflect out from the surface of the absorber. The area surrounded by white dashed lines represents impedance values between 0.8 and 1.2, which is considered as a good impendence matching range. Unfortunately, the high electrical conductivity of porous C/sp-SiOC-0 composites leads to impedance imbalance. The results show that the addition of PSA resin effectively adjusts the impedance matching value of porous carbon materials to gradually approach the white dotted area (shown in Figure 9).

To further compare the EM performances of porous C/sp-SiOC composite materials, Table 5 summarizes the microwave absorption performance of reported materials, including porous carbon-absorbing materials generated from pitch or other carbon sources [44,45,46,47,48,49,50]. It can be seen that the prepared C/sp-SiOC composites exhibit relatively high microwave absorption performance at a low thickness.

### 3.4. Absorption Mechanism of Porous C/sp-SiOC

Figure 10 further illustrates the possible microwave absorption mechanism of porous C/sp-SiOC-n (n = 4, 6, 8, and 12) composites. First, the addition of amorphous SiOC ceramics regulates the conductivity of carbon materials and improves impedance matching characteristics, which allows more incident waves to enter into the material instead of reflecting on the material surface. Second, this unique structure cannot construct an interconnected conductive network [51] so that the electromagnetic wave energy is converted to micro-current, resulting in strong conductive loss [52]. Moreover, these rich mesoporous structures would mainly produce many carbon–air interfaces, which would generate intense interfacial polarization. These porous structures also can provide multiple reflections and scattering channels of incident electromagnetic waves to further expand their propagation paths and gradually attenuate them [53]. There are a large number of interfaces between amorphous SiOC spheres and carbon, which leads to the accumulation of interface charges and electronic relaxation polarization [54].

## 4. Conclusions

In this work, porous C/sp-SiOC composites were successfully synthesized using pitch as a carbon precursor and PSA liquid as the precursor for SiOC spheres, through a simple melt-blending-phase separation route. This carbon/ceramic material transformed by the binary precursor provides a new idea for the preparation of a composite absorber. Moreover, both the carbon and the ceramic phase in the composite absorber are well combined, and there is no obvious separation of two phases. With the pitch as carbon source, this manufactured composite absorber become low cost and environmentally friendly. According to the above result, the as-prepared C/sp-SiOC composites have a unique structure, with the SiOC micro-particles generated in situ in the porous carbon. The porous C/sp-SiOC-8 composite exhibits excellent EM wave absorption performance. When the matching thickness was 1.39 mm, the minimum reflection loss was −56.85 dB, and its broadest adequate absorption bandwidth was up to 4.1 GHz (13.9 GHz-18 GHz) at 1.39 mm and wholly covers the X band. The superior microwave absorbing property of the porous C/sp-SiOC-n (n = 4, 6, 8, and 12) composites is due to their unique structure. The porous carbon skeleton may provide many paths for current micro transport, resulting in a conductive loss, whereas the addition of SiOC ceramics can adjust the dielectric constant to achieve a good impedance matching based on maintaining the original structure of porous carbon. More importantly, the investigation activation of this work provides a new idea for the pitch, which is instructive in developing durable and lightweight WMA materials. In particular, this carbon-based composite absorber is apt for the electromagnetic shielding shield layer for spacecraft or load-bearing and stealth components for stealth planes.

## Figures and Tables

**Figure 1 materials-15-08864-f001:**
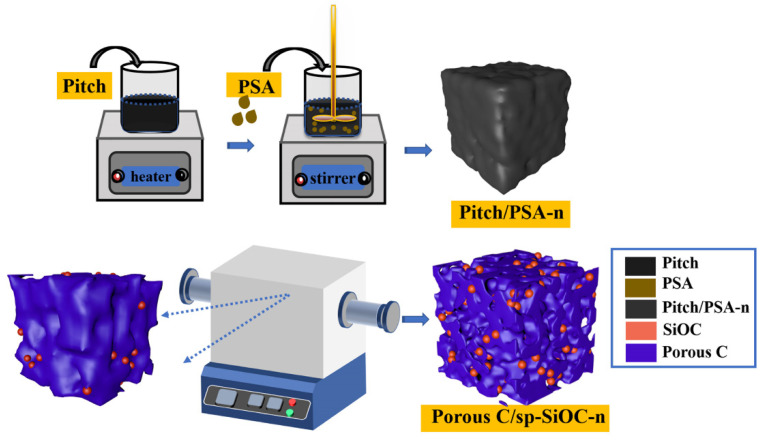
The schematic illustration of the synthesis of porous C/sp-SiOC composites.

**Figure 2 materials-15-08864-f002:**
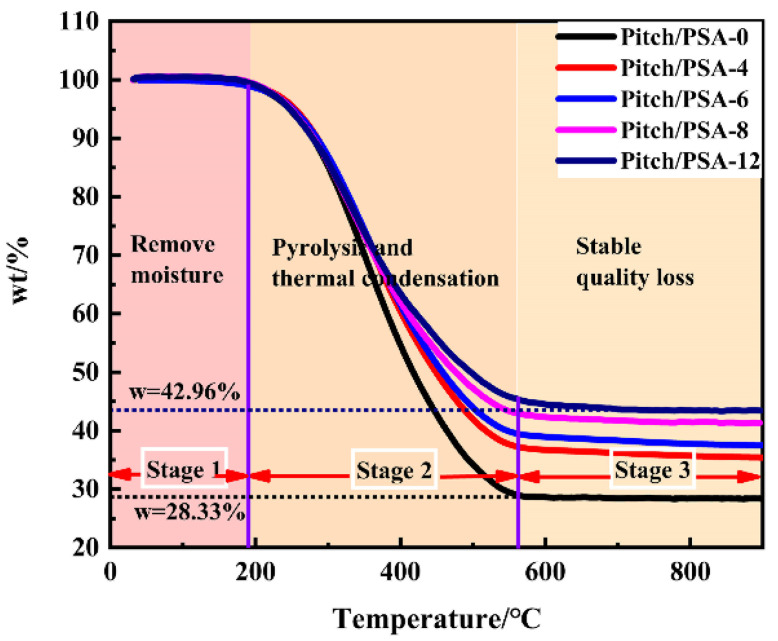
TG curves of samples prepared in laboratory.

**Figure 3 materials-15-08864-f003:**
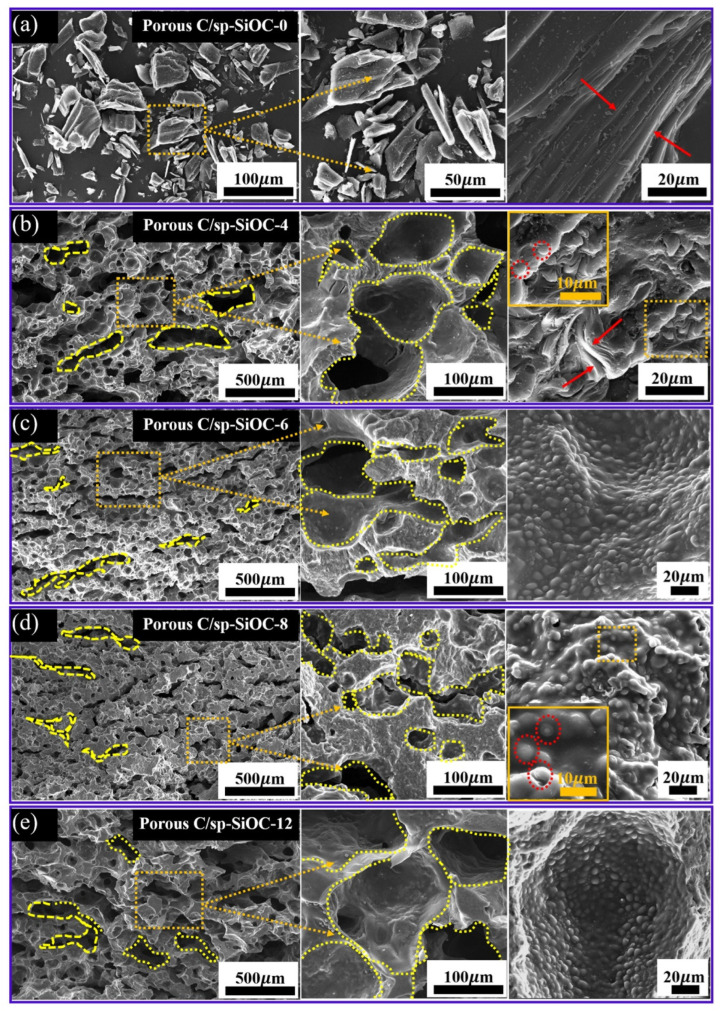
SEM images of the carbonization products at 1000 °C: (**a**) porous C/sp-SiOC-0 composite, (**b**) porous C/sp-SiOC-4 composite, (**c**) porous C/sp-SiOC-6 composite, (**d**) porous C/sp-SiOC-8 composite, (**e**) porous C/sp-SiOC-12 composite.

**Figure 4 materials-15-08864-f004:**
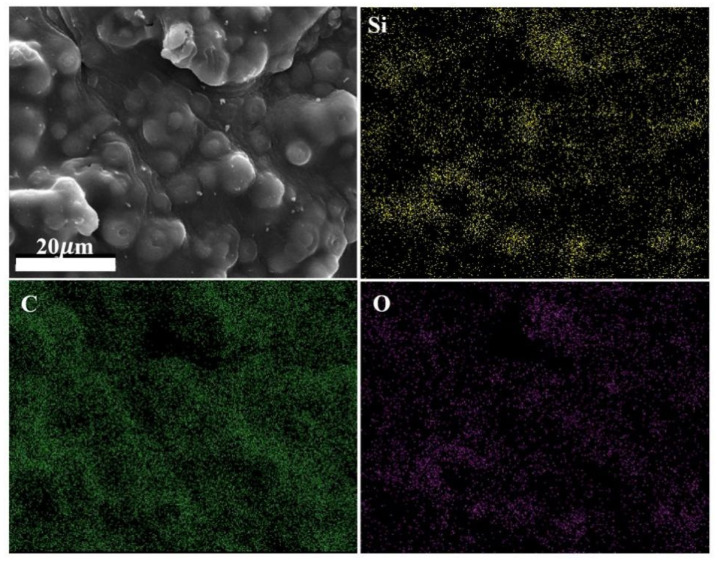
EDS images of the integral structure of SiOC spheres of the porous C/sp-SiOC-8 composite.

**Figure 5 materials-15-08864-f005:**
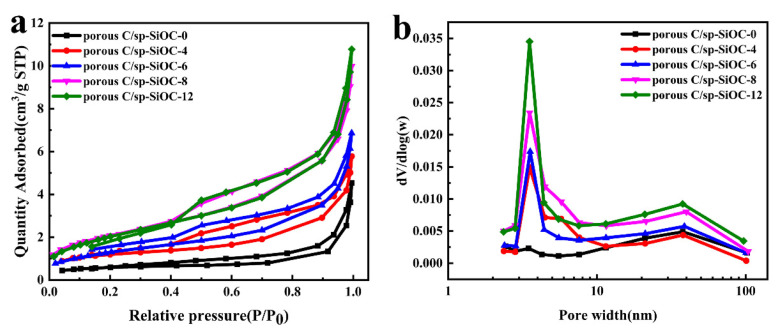
(**a**) Nitrogen adsorption–desorption isotherms. (**b**) Pore size distribution curves of all samples.

**Figure 6 materials-15-08864-f006:**
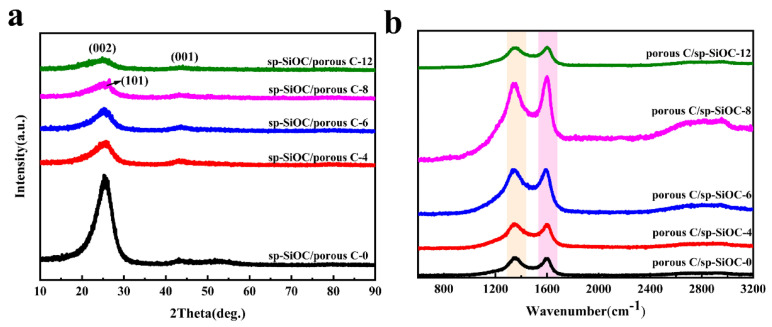
(**a**) XRD spectra of all samples. (**b**) Raman spectra of all as-received porous C/sp-SiOC composite showing the D and G band.

**Figure 7 materials-15-08864-f007:**
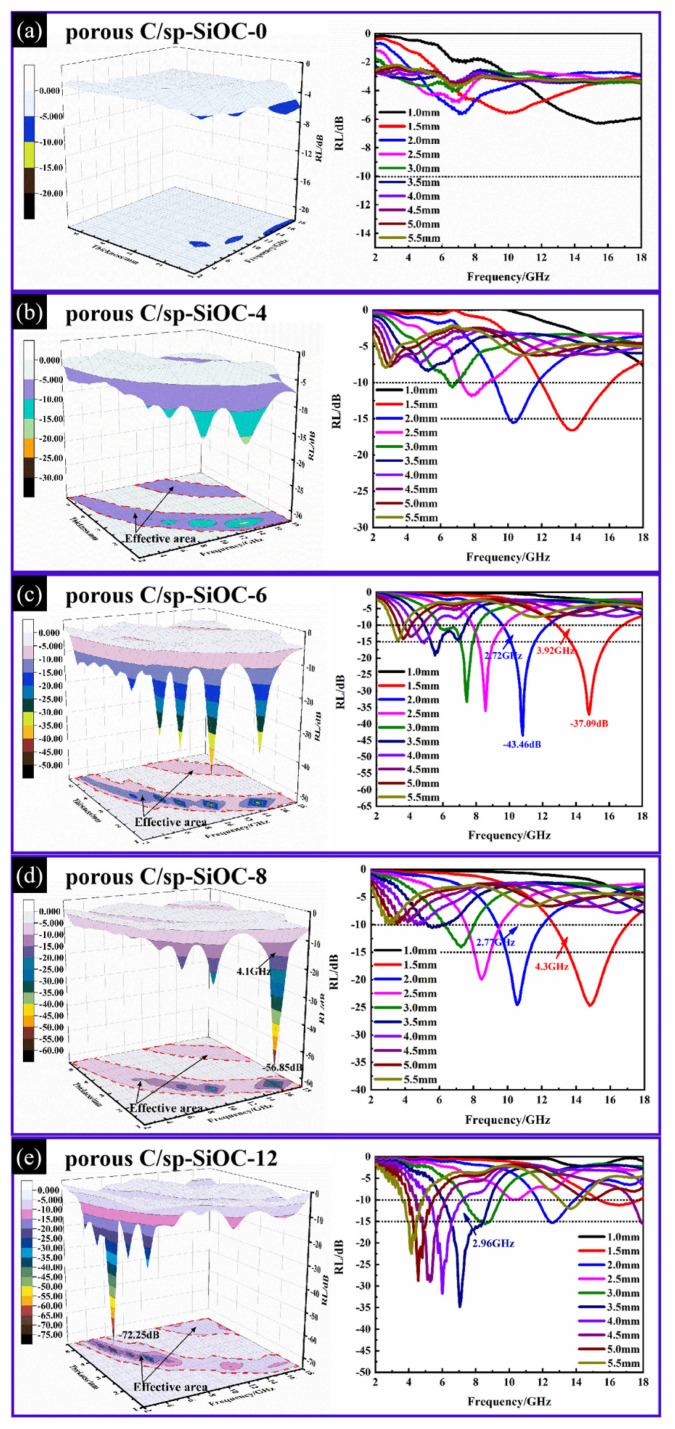
EM wave reflection loss of porous C/sp-SiOC-n (n = 0, 4, 6, 8, and 12) in the frequency range of 2–18 GHz.

**Figure 8 materials-15-08864-f008:**
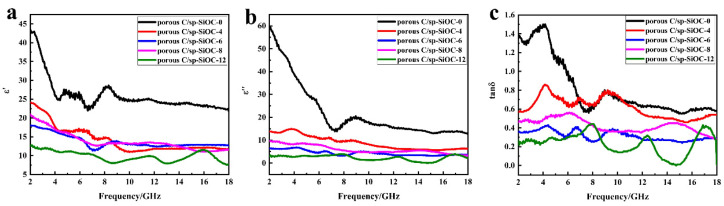
Relative complex permittivity of the composites: (**a**) the real part *ε′*, (**b**) the imaginary part *ε*″, (**c**) dielectric loss of permittivity tanδ.

**Figure 9 materials-15-08864-f009:**
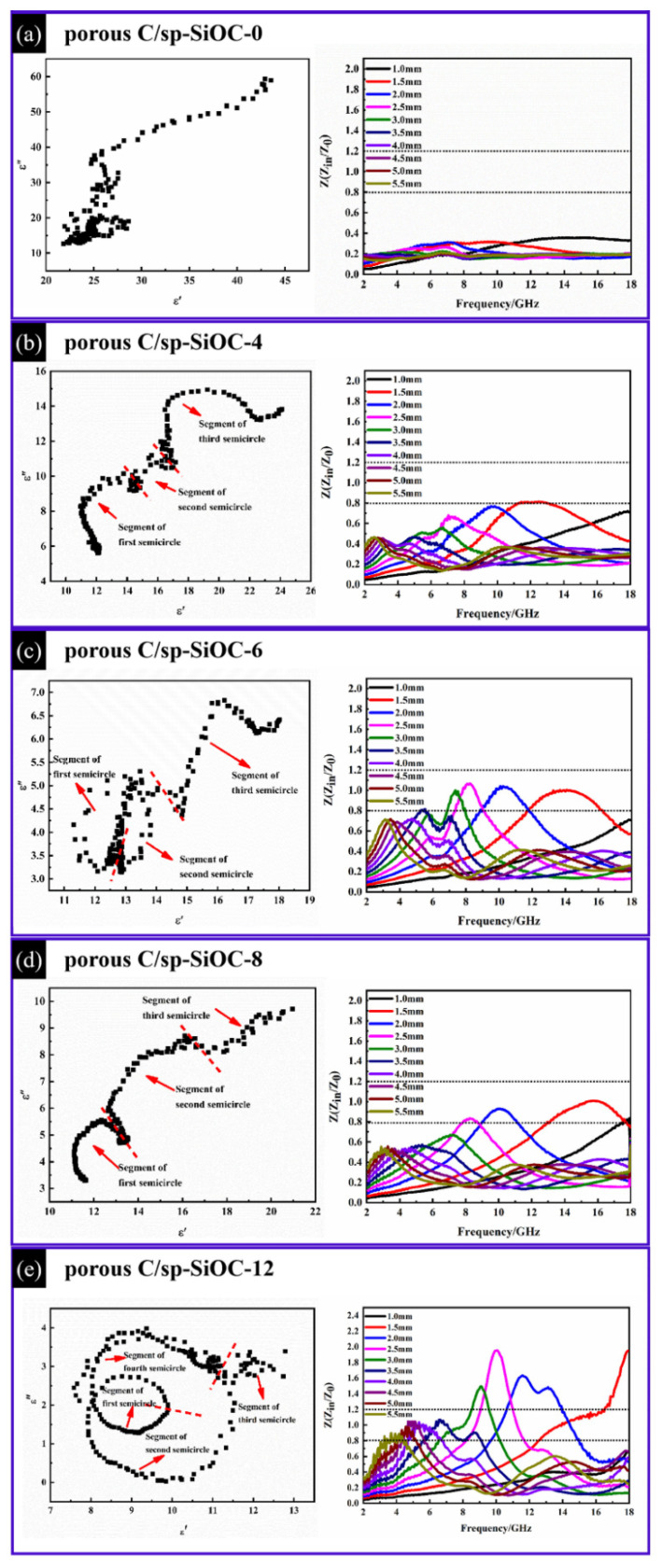
Cole–Cole semicircle curves and the impedance (Z) of the samples: (**a**) porous C/sp-SiOC-0, (**b**) porous C/sp-SiOC-4, (**c**) porous C/sp-SiOC-6, (**d**) porous C/sp-SiOC-8, (**e**) porous C/sp-SiOC-12.

**Figure 10 materials-15-08864-f010:**
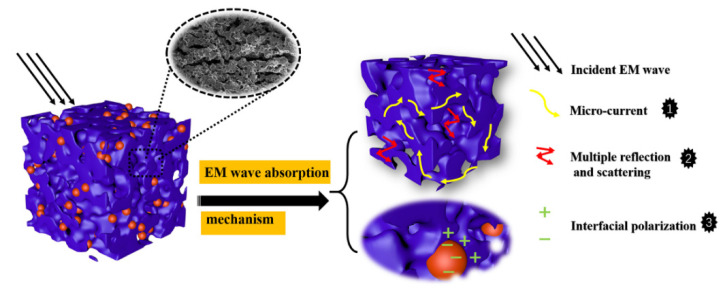
Schematic of the possible EM wave absorption mechanism of the porous C/sp-SiOC composite materials.

**Table 1 materials-15-08864-t001:** Proportion of components in slurry.

Samples	Precursor	Component		Final Weight (%)	Composition		Density (g/cm^3^)
		PSA (ml)	Pitch (g)		C (%)	SiOC (%)	
Porous C/sp-SiOC-0	Pitch/PSA-0	0	20	28.33	100	0	0.71
Porous C/sp-SiOC-4	Pitch/PSA-4	4		35.29	68	32	0.70
Porous C/sp-SiOC-6	Pitch/PSA-6	6		37.32	59	41	0.85
Porous C/sp-SiOC-8	Pitch/PSA-8	8		40.73	52	48	0.58
Porous C/sp-SiOC-12	Pitch/PSA-12	12		42.96	42	58	0.63
Porous C/sp-SiOC-14	Pitch/PSA-14	14					

**Table 2 materials-15-08864-t002:** Density and porosity of Cf/C-SiC composites with different binary binder content.

Samples	Binder Content (%)	Density (g·cm^−1^)	Open Porosity (%)
Cf/C-SiC-50%	50%	1.45	14.16
Cf/C-SiC-60%	60%	1.53	1.28
Cf/C-SiC-70%	70%	1.61	4.89

**Table 3 materials-15-08864-t003:** BET test data of samples.

Sample	BET Surface (m^2^g^−1^)	Pore Volume (cm^3^g^−1^)	Pore Size (nm)
Porous C/sp-SiOC-0	2.0591	0.006449	12.52667
Porous C/sp-SiOC-4	4.1852	0.008499	8.12340
Porous C/sp-SiOC-6	4.8935	0.010302	8.42063
Porous C/sp-SiOC-8	7.6335	0.015031	7.87405
Porous C/sp-SiOC-12	7.6288	0.016131	8.45788

**Table 4 materials-15-08864-t004:** Nomenclature of products obtained in the laboratory.

Samples	ID/IG	The Calculated Size of Graphite Microcrystalline
Porous C/sp-SiOC-0	1.07	4.11
Porous C/sp-SiOC-4	1.02	4.31
Porous C/sp-SiOC-6	1.01	4.36
Porous C/sp-SiOC-8	0.95	4.63
Porous C/sp-SiOC-12	0.98	4.49

**Table 5 materials-15-08864-t005:** Microwave absorption performance of the porous carbon-based composites.

Sample	Filling Ratio	EAB	Thickness	RL_min_	Ref.
Porous carbon	70 wt%	1.76 GHz	2 mm	−42.4 dB	[47]
PCNs-4	20 wt%	5.3 GHz	1.8 mm	−53.7 dB	[48]
Porous Ni/C	20 wt%	3.8 GHz	1.75 mm	−47 dB	[49]
S-FNGA	10 wt%	3.8 GHz	2.9 mm	−25.75 dB	[46]
FC500	70 wt%	4.8 GHz	1.5 mm	−31.05 dB	[50]
PC@PANI	20 wt%	6.64 GHz	2.6 mm	−72.16 dB	[45]
Porous carbon/Ni	33.3 wt%	5.46 GHz	1.7 mm	−42.51 dB	[44]
Porous C/sp-SiOC	50 wt%	4.1 GHz	1.39 mm	−56.85 dB	This work

## Data Availability

Not applicable.

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
