# Peer review of "Enhanced Electromagnetic Wave Absorption of SiOC/Porous Carbon Composites"

_materials, 2022, doi:10.3390/ma15248864_

Round 1

Reviewer 1 Report

In this work, the authors studied a new absorbent material with a unique structure, which could combine the advantages of porous carbon and ceramic materials. This composite material with SiOC ceramic spheres embedded in a porous 3D carbon structure was designed and prepared using a polymer-derived method using pitch and PSA, which were precursors of carbon and Si-O-C. They studied, in detail, the phase structure and micromorphology of composite materials. The absorption mechanisms behind the excellent microwave absorption capacity were also discussed.

Congratulations. The work is good and well structured, however, there are some concerns about your work that can be resolved by improving the work and your understanding. Please see the attachment.

Author Response

Response to Reviewers 1

Materials Paper No. Materials-2027994

Title: SiOC micro-particles embedded porous carbon composite pre-pared by precursor conversion method with enhanced EM absorbing performance

Authors: Wen Yang et al.

We are grateful to the editors and all reviewers for editing and reviewing our manuscript, and for these useful and constructive comments. We have revised the manuscript based on the provided comments and suggestions. Our responses to the referee are below-please note that the reviewers’ comments are given in bold, followed by our responses are given in colorful fonts. Also, the added parts in the revised manuscript are highlighted in red and as underlined text.

Reviewer #1:

Comments to Author:

In this work, the authors studied a new absorbent material with a unique structure, which could combine the advantages of porous carbon and ceramic materials. This composite material with SiOC ceramic spheres embedded in a porous 3D carbon structure was designed and prepared using a polymer-derived method using pitch and PSA, which were precursors of carbon and Si-O-C. They studied, in detail, the phase structure and micromorphology of composite materials. The absorption mechanisms behind the excellent microwave absorption capacity were also discussed. Congratulations. The work is good and well structured, however, there are some concerns about your work that can be resolved by improving the work and your understanding, such as:

Query (1) Write the abbreviation CF before its meaning. Carbon Fiber (CF). Check if there are more abbreviations without being written in full.

Response: Thanks for the Reviewer’s valuable comments and suggestion.

The full name (Carbon Fiber) of CF in this paper have been added (Page 2, line 59), and all the abbreviations in this paper have been checked.

Query (2) The sentence from lines 65 and 66: “However, due to the poor adhesion between carbon and ceramic components, its use as microwave absorbing materials is greatly limited.”, should have a reference that supports the authors' assertion.

Response: Thanks for the Reviewer’s valuable comments and suggestion.

To better express the meaning, the sentence has revised to “However, due to the poor combination and distribution between carbon and ceramic components, its use as microwave absorbing materials is greatly limited [13].” The supporting reference [13] has added in this sentence (Page 2, line 65-66, and Page 16, line 443-444)

  1. Mao, B.; Xia, X.; Qin, R.; Xu, D.; Wang, X.; Lv, H. Synthesis and microwave absorption properties of multilayer SiC/C foam with alternating distribution of C and SiC. J. Alloys Compd 2021, 879, 160440.

Query (3) In line 94. The authors speak of “our laboratory”. The first person of the plural is used. Please avoid this situation, because in scientific documents the first person does not be used. Please, check the words “our”, “we”, “us”.

Response: Thanks for the Reviewer’s valuable comments and suggestion.

The sentence “The polysilylacetylene (PSA) used in this work was synthesized in our lab” in line 94 has been revised to “The polysilylacetylene (PSA) used in this work was synthesized in the lab, and the detaild process was described in the previous work.” (Page 2, line 93-94). In addition, the other first person sentences are all corrected in the paper.

Query (4) In line 268: “In addition, we plot the microwave absorption properties…”, fix the sentence.

Response: Thanks for the Reviewer’s valuable comments and suggestion.

The sentence “In addition, we plot the microwave absorption properties of the samples under different thicknesses.” in line 268 has been revised to “In addition, the microwave absorption properties of samples at different thicknesses exhibit an obvious changes and are clearly reflected in the results .” (Page 11, line 271-272).

Query (5) In sub-chapter 2.3, the authors talk about Scanning Electron Microscopy and Energy Dispersive Spectroscopy. The identification of the machines used is missing. Check if there are more machines to be identified.

Response: Thanks for the Reviewer’s valuable comments and suggestion.

The identification of the Scanning Electron Microscopy and Energy Dispersive Spectroscopy are added in chapter 2.3, and the identification other machines are also added. The chapter 2.3 is revised to “The scanning electron microscope (SEM, Zeiss Merlin compact scanning electron microscope) was used to test the microstructure of the samples, and the elements of the internal structure were tested by Energy Dispersive Spectrometer (EDS, Bruker Quantax Xflash 60 SDD). The surface area and pore diameter measurements of porous C/sp-SiOC composites were measured by Brunner-Emmet-Teller (BET, Micromeritics ASAP) equipment with nitrogen gas adsorption-desorption at -196℃. . X-ray diffractometer (XRD, Rigaku XRD 2500, Cu Kα, 40.0 KV, 30.0 mA) was identified the phase structures of composite materials. The thermo-stability of samples was obtained by using a thermogravimetric analysis (TG, PerkinElmer Diamond) apparatus with a temperature of 10℃ /min in a nitrogen atmosphere. The chemical structures of samples were obtained using a Fourier transform infrared spectrometer (FT-IR, Thermo Nicolet AVATAR 370) with a resolution of 0.2 cm-1 in the range of 400-4000 cm-1, respectively.”(Page 4, line 113-124)

Query (6) In line 342, the authors refer to table 5. The work does not have table 4 or table 5. Check the error and change it.

Response: Thanks for the Reviewer’s valuable comments and suggestion.

The missing table 4 and the related references (Ref 45-50) have added in the text (Page 12, line 346, 351-353, Page17, 510-522), and the microwave absorption performances of the different porous carbon-based composites has listed in this table. The table 4 and the related references (Ref 45-50) is shown as follows:

Table 4. Microwave absorption performances of the porous carbon-based composites.

sample

Filling ratio

EAB

Thickness

RLmin

Ref.

Porous carbon

70 wt%

1.76 GHz

2 mm

-42.4 dB

[47]

PCNs-4

20 wt%

5.3 GHz

1.8 mm

-53.7 dB

[48]

Porous Ni/C

20 wt%

3.8 GHz

1.75 mm

-47 dB

[49]

S-FNGA

10 wt%

3.8 GHz

2.9 mm

-25.75 dB

[46]

FC500

70 wt%

4.8 GHz

1.5 mm

-31.05 dB

[50]

PC@PANI

20 wt%

6.64 GHz

2.6 mm

-72.16 dB

[45]

Porous carbon/Ni

33.3 wt%

5.46 GHz

1.7 mm

-42.51 dB

[44]

porous C/sp-SiOC

50 wt%

4.1 GHz

1.39 mm

-56.85 dB

This work

  1. Zhang, F.; Cui, W.; Wang, B.; Xu, B.; Liu, X.; Liu, X.; Jia, Z.; Wu, G. Morphology-control synthesis of polyaniline decorative porous carbon with remarkable electromagnetic wave absorption capabilities. Composites Part B: Engineering 2021, 204, 108491.
  2. Liu, W.; Tan, S.; Yang, Z.; Ji, G. Hollow graphite spheres embedded in porous amorphous carbon matrix as lightweight and low-frequency microwave absorbing material through modulating dielectric loss. Carbon 2018, 138, 143-153.
  3. Qiu, X.; Wang, L.; Zhu, H.; Guan, Y.; Zhang, Q. Lightweight and efficient microwave absorbing materials based on walnut shell-derived nano-porous carbon[J]. Nanoscale 2017, 9(22): 7408-7418.
  4. Yang, W.; Li, R.; Jiang, B.; Wang, T.; Hou, L.; Li, Z.; Liu, Z.; Yang, F.; Li, Y. Production of hierarchical porous carbon nanosheets from cheap petroleum asphalt toward lightweight and high-performance electromagnetic wave absorbents. Carbon 2020, 166, 218-226.
  5. Liu, C.; Lin, Z.; Chen, C.; Kirk, D; Xu, Y. Porous C/Ni composites derived from fluid coke for ultra-wide bandwidth elec-tromagnetic wave absorption performance. Chem. Eng. J. 2019, 366, 415-422.
  6. Wang, L.; Guan, Y.; Qiu, X.; Zhu, H.; Pan, S.; Yu, M.; Zhang, Q. Efficient ferrite/Co/porous carbon microwave absorbing ma-terial based on ferrite@metal–organic framework. Chem. Eng. J. 2017, 326, 945-955.

Query (7) Be careful with formatting. It has some errors. Check all work , for example, missing spaces between the words and the square brackets of the references. (Line 37: waves[1,2]; Line 40; Carbon fiber[4]…)

Response: Thanks for the Reviewer’s valuable comments and suggestion.

The missing spaces between the words and the square brackets in the references have been added (Page 1 Line 37 and Line 40 ), and the same incorrect formatting in the text have been corrected in this full paper.

Query (8) The conclusions would be richer if the authors presented comparative values with results from other authors, emphasizing the added value of this work compared to what is currently used.

Response: Thanks for the Reviewer’s valuable comments and suggestion.

The innovation and application both are both introduced in the conclusion (Page 15, line 378-384, line 391-396). The content of conclusion have been updated as“In this work, the porous C/sp-SiOC composites were successfully synthesized using the pitch as a carbon precursor, the PSA liquid as the precursor for SiOC spheres, through a simple melt-blending-phase separation route. This carbon/ceramic material transformed by the binary precursor provides a new idea for the preparation of composite absorber. Moreover, both of the carbon and ceramic phase in the composite absorber are well com-bined, while there is no obvious separation of two phases. With the pitch as carbon source, this manufactured of composite absorber become low cost and environmentally-friendly. According to the above result, the as prepared C/sp-SiOC composites have an unique structure, the SiOC micro-particles generated in situ in the porous carbon. The porous C/sp-SiOC-8 composite exhibits excellent EM wave absorption performance. When the matching thickness is 1.39 mm, the minimum reflection loss is -56.85 dB, and its broadest adequate absorption bandwidth was up to 4.1 GHz (13.9 GHz-18 GHz) at 1.39 mm covers the X band practically wholly. The superior microwave absorbing property of the porous C/sp-SiOC-n (n=4, 6, 8, 12) composites is due to their unique structure. The porous carbon skeleton may provide many paths for current micro transport, resulting in a conductive loss, while the addition of SiOC ceramics can adjust the dielectric constant to achieve a good impedance matching based on maintaining the original structure of porous carbon. More importantly, the investigation activation of this work provides a new idea for the pitch, which is instructive in developing durable and lightweight WMA materials. In par-ticular, this carbon-based composite absorber is apt to the electromagnetic shielding shield layer for spacecraft or Load-bearing and stealth components for stealth plane.”

Reviewer 2 Report

The paper is nice and well written, and reports on an important topic. I just have some minor suggestions:

The english is understandable but could use some corrections from a native english speaker or someone with adequate skills. For example,

line 18: "which took the low cost pitch as carbon resource" would be better if it was "which used the low cost pitch as carbon resource",

lines 33 and 34, the work "equipment" appears twice in the same sentence. Instead of "computer equipment" just say "computers", for example.

line 54, some strange commas are found there.

Do not use abbreviations in the title (instead of EM write eletromagnetic).

Also the title is too long. Is it really needed to say in the title that composites were "prepared by precursor conversion method". By the way, what is precursor conversion? Do you mean precursor decomposition?

Authors also should better state what is the novelty of the paper.

Also authors should say what was the motivation to use pitch as a carbon source. 

A comparison of the absorbing properties of the reported composites with other materials from literature and their absorbing properties should be provided, in order to better evaluate the samples presented in this study.

Author Response

Response to Reviewers 2

Materials Paper No. Materials-2027994

Title: SiOC micro-particles embedded porous carbon composite pre-pared by precursor conversion method with enhanced EM absorbing performance

Authors: Wen Yang et al.

We are grateful to the editors and all reviewers for editing and reviewing our manuscript, and for these useful and constructive comments. We have revised the manuscript based on the provided comments and suggestions. Our responses to the referee are below-please note that the reviewers’ comments are given in bold, followed by our responses are given in colorful fonts. Also, the added parts in the revised manuscript are highlighted in red and as underlined text.

Reviewer #2:

Comments to Author:

The paper is nice and well written, and reports on an important topic. I just have some minor suggestions:

Query (1) The English is understandable but could use some corrections from a native English speaker or someone with adequate skills. For example,

line 18: "which took the low cost pitch as carbon resource" would be better if it was "which used the low cost pitch as carbon resource",

lines 33 and 34, the work "equipment" appears twice in the same sentence. Instead of "computer equipment" just say "computers", for example.

line 54, some strange commas are found there.

Response: Thanks for the Reviewer’s valuable comments and suggestion.

The mentioned sentence have corrected as follows and the same sentences are all checked in the paper.

The sentence “which took the low cost pitch as carbon resource” has been revised as “which used the low cost pitch as carbon resource” (Page 1, line 17-18).

The sentence “The emergence and broad application of electronic equipment (telephones, computer equipment, and base station, etc.) have caused serious microwave radiation and microwave pollution” has been revised as “The emergence and broad application of electronic equipment (telephones, computer, and base station, etc.) have caused serious microwave radiation and microwave pollution” (Page 1, line 32-33).

The sentence “Fortunately, the research shows that the introduction of ceramic components (SiC[13]、Si-C-N[14]、Si-B-C-N[15], et al.) into pure carbon materials can effectively adjust the electromagnetic parameters, balance the impedance matching conditions and improve the electromagnetic wave absorption performance of materials

” has been revised as “Fortunately, the introduction of ceramic elements into pure carbon materials is beneficial to improve the electromagnetic absorption performance, such as the SiC[13]、Si-C-N [14]、Si-B-C-N [15], et al. These composites materials can effectively adjust the electromagnetic parameters, balance the impedance matching conditions and improve the electromagnetic wave absorption performance of materials.” (Page 2, line 52-56).

Query (2) Do not use abbreviations in the title (instead of EM write eletromagnetic).

Also the title is too long. Is it really needed to say in the title that composites were "prepared by precursor conversion method". By the way, what is precursor conversion? Do you mean precursor decomposition?

Response: Thanks for the Reviewer’s valuable comments and suggestion.

The title “SiOC micro-particles embedded porous carbon composite pre-pared by precursor conversion method with enhanced EM absorbing performance”is revised as “Enhanced electromagnetic wave absorption of SiOC/porous carbon composites”(Page 1 line 2-3). The precursor conversion in the title mean the decomposition of precursor, which usually described as PDC (polymer or precursor derived ceramics)

Query (3) Authors also should better state what is the novelty of the paper.

Response: Thanks for the Reviewer’s valuable comments and suggestion.

The novelty of the paper are added in the conclusion as follows: “In this work, the porous C/sp-SiOC composites were successfully synthesized using the pitch as a carbon precursor, the PSA liquid as the precursor for SiOC spheres, through a simple melt-blending-phase separation route. This carbon/ceramic material transformed by the binary precursor provides a new idea for the preparation of composite absorber. Moreover, both of the carbon and ceramic phase in the composite absorber are well com-bined, while there is no obvious separation of two phases. With the pitch as carbon source, this manufactured of composite absorber become low cost and environmentally-friendly. According to the above result, the as prepared C/sp-SiOC composites have an unique structure, the SiOC micro-particles generated in situ in the porous carbon.”(Page 15, 378-384)

Query (4) Also authors should say what was the motivation to use pitch as a carbon source. 

Response: Thanks for the Reviewer’s valuable comments and suggestion.

The pitch is usually used as a precursor for carbon/carbon composites, which could convert to carbon matrix during pyrolysis process. In this paper, the pitch and polysilylacetylene were used as the precursor to obtain a SiOC/C composites. Because of its thermoplastic, the pitch could combine with polysilylacetylene evently during the melt-blending route, which is benefit to control the distribution between carbon and ceramic components.

Query (5) A comparison of the absorbing properties of the reported composites with other materials from literature and their absorbing properties should be provided, in order to better evaluate the samples presented in this study.

Response: Thanks for the Reviewer’s valuable comments and suggestion.

The microwave absorption performances of the different porous carbon-based composites has listed in this newly added table 4 (Page 12, line 351-353, Page17, 510-522). According to the table 4, the good absorbing properties in this work has exhibited. The table 4 and the related references (Ref 45-50) is shown as follows:

Table 4. Microwave absorption performances of the porous carbon-based composites.

sample

Filling ratio

EAB

Thickness

RLmin

Ref.

Porous carbon

70 wt%

1.76 GHz

2 mm

-42.4 dB

[47]

PCNs-4

20 wt%

5.3 GHz

1.8 mm

-53.7 dB

[48]

Porous Ni/C

20 wt%

3.8 GHz

1.75 mm

-47 dB

[49]

S-FNGA

10 wt%

3.8 GHz

2.9 mm

-25.75 dB

[46]

FC500

70 wt%

4.8 GHz

1.5 mm

-31.05 dB

[50]

PC@PANI

20 wt%

6.64 GHz

2.6 mm

-72.16 dB

[45]

Porous carbon/Ni

33.3 wt%

5.46 GHz

1.7 mm

-42.51 dB

[44]

porous C/sp-SiOC

50 wt%

4.1 GHz

1.39 mm

-56.85 dB

This work

  1. Zhang, F.; Cui, W.; Wang, B.; Xu, B.; Liu, X.; Liu, X.; Jia, Z.; Wu, G. Morphology-control synthesis of polyaniline decorative porous carbon with remarkable electromagnetic wave absorption capabilities. Composites Part B: Engineering 2021, 204, 108491.
  2. Liu, W.; Tan, S.; Yang, Z.; Ji, G. Hollow graphite spheres embedded in porous amorphous carbon matrix as lightweight and low-frequency microwave absorbing material through modulating dielectric loss. Carbon 2018, 138, 143-153.
  3. Qiu, X.; Wang, L.; Zhu, H.; Guan, Y.; Zhang, Q. Lightweight and efficient microwave absorbing materials based on walnut shell-derived nano-porous carbon[J]. Nanoscale 2017, 9(22): 7408-7418.
  4. Yang, W.; Li, R.; Jiang, B.; Wang, T.; Hou, L.; Li, Z.; Liu, Z.; Yang, F.; Li, Y. Production of hierarchical porous carbon nanosheets from cheap petroleum asphalt toward lightweight and high-performance electromagnetic wave absorbents. Carbon 2020, 166, 218-226.
  5. Liu, C.; Lin, Z.; Chen, C.; Kirk, D; Xu, Y. Porous C/Ni composites derived from fluid coke for ultra-wide bandwidth elec-tromagnetic wave absorption performance. Chem. Eng. J. 2019, 366, 415-422.
  6. Wang, L.; Guan, Y.; Qiu, X.; Zhu, H.; Pan, S.; Yu, M.; Zhang, Q. Efficient ferrite/Co/porous carbon microwave absorbing ma-terial based on ferrite@metal–organic framework. Chem. Eng. J. 2017, 326, 945-955.

Reviewer 3 Report

This paper reports on the SiOC micro-particles embedded porous carbon composite prepared by precursor conversion method with enhanced EM absorbing performance”. Introduction and conclusion, methodology and reference, results and discussion seems be corrected.

I have few comments to the manuscript:

1.     All manuscript similar correction. Add the missing space “microscope(SEM)” to “microscope (SEM)”.

2.     Extend the description of the research carried out, describe it more clearly, add references.

3.     In the conclusions, describe the application for which the obtained materials can be used.

Taking into account all comments the manuscript may be published in Materials after minor revision.

Author Response

Response to Reviewers 3

Materials Paper No. Materials-2027994

Title: SiOC micro-particles embedded porous carbon composite pre-pared by precursor conversion method with enhanced EM absorbing performance

Authors: Wen Yang et al.

We are grateful to the editors and all reviewers for editing and reviewing our manuscript, and for these useful and constructive comments. We have revised the manuscript based on the provided comments and suggestions. Our responses to the referee are below-please note that the reviewers’ comments are given in bold, followed by our responses are given in colorful fonts. Also, the added parts in the revised manuscript are highlighted in red and as underlined text.

Reviewer #3:

Comments to Author:

This paper reports on the “SiOC micro-particles embedded porous carbon composite prepared by precursor conversion method with enhanced EM absorbing performance”. Introduction and conclusion, methodology and reference, results and discussion seems be corrected.

I have few comments to the manuscript:

Query (1) All manuscript similar correction. Add the missing space “microscope(SEM)” to “microscope (SEM)”.

Response: Thanks for the Reviewer’s valuable comments and suggestion.

The missing spaces between the words and the square brackets in the references have been added as follows (Page 4, line 113-124), and the same incorrect formatting in the text have been corrected in this full paper.

The chapter 2.3 is revised to “The scanning electron microscope (SEM, Zeiss Merlin compact scanning electron microscope) was used to test the microstructure of the samples, and the elements of the internal structure were tested by Energy Dispersive Spectrometer (EDS, Bruker Quantax Xflash 60 SDD). The surface area and pore diameter measurements of porous C/sp-SiOC composites were measured by Brunner-Emmet-Teller (BET, Micromeritics ASAP) equipment with nitrogen gas adsorption-desorption at -196℃. . X-ray diffractometer (XRD, Rigaku XRD 2500, Cu Kα, 40.0 KV, 30.0 mA) was identified the phase structures of composite materials. The thermo-stability of samples was obtained by using a thermogravimetric analysis (TG, PerkinElmer Diamond) apparatus with a temperature of 10℃ /min in a nitrogen atmosphere. The chemical structures of samples were obtained using a Fourier transform infrared spectrometer (FT-IR, Thermo Nicolet AVATAR 370) with a resolution of 0.2 cm-1 in the range of 400-4000 cm-1, respectively.”(Page 4, line 113-124)

Query (2) Extend the description of the research carried out, describe it more clearly, add references.

Response: Thanks for the Reviewer’s valuable comments and suggestion.

The detailed description of the research has added in the conclusion (the detailed content in conclusion is listed in the response to query 3). To better understand this text, some related reference ([45-50]) have added in this paper as follows:

  1. Zhang, F.; Cui, W.; Wang, B.; Xu, B.; Liu, X.; Liu, X.; Jia, Z.; Wu, G. Morphology-control synthesis of polyaniline decorative porous carbon with remarkable electromagnetic wave absorption capabilities. Composites Part B: Engineering 2021, 879, 108491.
  2. Liu, W.; Tan, S.; Yang, Z.; Ji, G. Hollow graphite spheres embedded in porous amorphous carbon matrix as lightweight and low-frequency microwave absorbing material through modulating dielectric loss. Carbon 2018, 138, 143-153.
  3. Qiu, X.; Wang, L.; Zhu, H.; Guan, Y.; Zhang, Q. Lightweight and efficient microwave absorbing materials based on walnut shell-derived nano-porous carbon[J]. Nanoscale 2017, 9(22): 7408-7418.
  4. Yang, W.; Li, R.; Jiang, B.; Wang, T.; Hou, L.; Li, Z.; Liu, Z.; Yang, F.; Li, Y. Production of hierarchical porous carbon nanosheets from cheap petroleum asphalt toward lightweight and high-performance electromagnetic wave absorbents. Carbon 2020, 166, 218-226.
  5. Liu, C.; Lin, Z.; Chen, C.; Kirk, D; Xu, Y. Porous C/Ni composites derived from fluid coke for ultra-wide bandwidth elec-tromagnetic wave absorption performance. Chem. Eng. J. 2019, 366, 415-422.
  6. Wang, L.; Guan, Y.; Qiu, X.; Zhu, H.; Pan, S.; Yu, M.; Zhang, Q. Efficient ferrite/Co/porous carbon microwave absorbing ma-terial based on ferrite@metal–organic framework. Chem. Eng. J. 2017, 326, 945-955.

Query (3) In the conclusions, describe the application for which the obtained materials can be used.

Response: Thanks for the Reviewer’s valuable comments and suggestion.

The innovation and application both are both introduced in the conclusion (Page 15, line 378-384, line 391-396). The content of conclusion have been updated as“In this work, the porous C/sp-SiOC composites were successfully synthesized using the pitch as a carbon precursor, the PSA liquid as the precursor for SiOC spheres, through a simple melt-blending-phase separation route. This carbon/ceramic material transformed by the binary precursor provides a new idea for the preparation of composite absorber. Moreover, both of the carbon and ceramic phase in the composite absorber are well com-bined, while there is no obvious separation of two phases. With the pitch as carbon source, this manufactured of composite absorber become low cost and environmentally-friendly. According to the above result, the as prepared C/sp-SiOC composites have an unique structure, the SiOC micro-particles generated in situ in the porous carbon. The porous C/sp-SiOC-8 composite exhibits excellent EM wave absorption performance. When the matching thickness is 1.39 mm, the minimum reflection loss is -56.85 dB, and its broadest adequate absorption bandwidth was up to 4.1 GHz (13.9 GHz-18 GHz) at 1.39 mm covers the X band practically wholly. The superior microwave absorbing property of the porous C/sp-SiOC-n (n=4, 6, 8, 12) composites is due to their unique structure. The porous carbon skeleton may provide many paths for current micro transport, resulting in a conductive loss, while the addition of SiOC ceramics can adjust the dielectric constant to achieve a good impedance matching based on maintaining the original structure of porous carbon. More importantly, the investigation activation of this work provides a new idea for the pitch, which is instructive in developing durable and lightweight WMA materials. In par-ticular, this carbon-based composite absorber is apt to the electromagnetic shielding shield layer for spacecraft or Load-bearing and stealth components for stealth plane.”
